# Green Synthesis of ZnO/BC Nanohybrid for Fast and Sensitive Detection of Bisphenol A in Water

**Jiafeng Hu, Dongpeng Mao, Penghu Duan, Kelan Li, Yuqing Lin, Xinyao Wang and Yunxian Piao ***

Key Laboratory of Groundwater Resources and Environment, Ministry of Education,
Jilin Provincial Key Laboratory of Water Resources and Environment, College of New Energy and Environment,
Jilin University, Changchun 130021, China; hujf19@mails.jlu.edu.cn (J.H.); maodp20@mails.jlu.edu.cn (D.M.);
duanph20@mails.jlu.edu.cn (P.D.); likl19@mails.jlu.edu.cn (K.L.); linyq19@mails.jlu.edu.cn (Y.L.);
xinyaow20@mails.jlu.edu.cn (X.W.)
*** Correspondence: yxpiao@jlu.edu.cn; Tel.: +86-186-0441-4674

**Abstract:** A nanohybrid of zinc oxide and biochar (ZnO/BC) with high conductivity was green synthesized using a simple hydrothermal method, and utilized for the sensitive detection of bisphenol A (BPA) by coating the nanohybrid film on an electrode of glassy carbon. The ZnO/BC presented greatly improved electrocatalytic performance and electron transfer ability compared to the zinc oxide and biochar. The ZnO/BC film-coated electrode could detect the BPA in aqueous solution within 3 min while neglected interference from higher concentrations of regularly existing ions and similar concentrations of estradiol (E2), phenol, dichlorophenol (DCP), and ethinylestradiol (EE2). Under optimal conditions, the linear range of BPA detection was $5 \times 10^{-7} \sim 1 \times 10^{-4}$ mol/L, with a detection limit of $1 \times 10^{-7}$ mol/L, and the detection sensitivity was 92 mA/M. In addition, the ZnO/BC electrode could detect BPA in a real water sample with good signal recovery. This electrode, with the advantages of an easy preparation, low cost, and fast response time, could be potentially applicable for environmental monitoring.

**Keywords:** biochar; zinc oxide; bisphenol A; electrochemical sensor

## 1. Introduction

Bisphenol A (2, 2-bis(4-hydroxyphenyl)propane) is commonly used as a raw material for synthesizing industrial supplies. It makes plastic products transparent, colorless, and more durable [1,2]. The manufacture and wide use of products containing BPA have led to the widespread distribution of BPA in the environment, which causes long-term effects. As a type of endocrine disruptor, BPA can indirectly affect the normal function of the endocrine system, causing health problems, such as malformations, reproductive disorders, and an increased risk of tumors [3,4]. Therefore, it is necessary to establish an accurate and rapid identification method to ensure that the living environment is safe and healthy.

Currently, various methods are used for BPA detection, including high performance liquid chromatography (HPLC) [5], gas chromatography-mass spectrometry (GC-MS) [6], enzyme-linked immunoassay (ELISA) [7] and electrochemical method [8]. However, considering the complexity of the operation, length of detection time, equipment cost, and ability of real-time detection, the electrochemical method is known to be most suitable. It has the advantages of miniaturization, low cost, and rapid detection [9,10]. The electrochemical detection method usually converts the change of electrode surface into electrical signals, and then uses the electrochemical workstation for identification

There are many ways to improve the performance of electrochemical sensors. For instance, Li et al. in situ generated graphene structures and some carbonaceous fragments on the electrode surface by polarizing the electrodes. Significantly enhanced electrochemical performance and more edge plane sites/defects were generated by cyclic voltammetry [11], but the range of the scanning potential was very wide, and the carbonaceous substances

produced originally from the electrode itself would cause great damage to the electrode. Electrochemical sensors, modified by biomaterials, such as enzymes, antibodies, and nucleic acid, although they have high selectivity and sensitivity, may tend to have bottlenecks in terms of stability, and relatively longer preparation or detection times [12]. Similarly, aptasensors, the target recognition of which mainly depends on structural changes of nucleic acid strands [13], may easily damage the nucleic acid structure during preparation or detection. All of this may lead to false detection results [14]. Instead, the electrodes prepared with stable, cost effective and conductive nanomaterials are known to be sensitive and economically advantageous. Usually, highly sensitive detection can be achieved by the incorporation of greatly conductive materials when constructing electrodes [15–17]. Recently, graphene oxide, nanotubes, and other carbon materials have been reported to improve detection performances due to their high electrical conductivity, great active surface area, and fast electron transfer behavior [18–20]. Nonetheless, the search for a nanocarbon material with a wide source, low cost, and high conductivity remains one of the great challenges in the electrochemical sensor field.

In this study, a nanohybrid of ZnO/BC was green synthesized using a hydrothermal method and utilized to construct electrochemical sensors for the sensitive detection of BPA, and have the advantages of an easy preparation, high sensitivity, and low cost through the use of ubiquitous biochar (BC) as the raw material. BC is a highly functionalized carbon-rich material obtained by the pyrolysis or carbonization of biomass materials in an atmosphere with a low-oxygen concentration or under anoxic conditions. It has an amorphous atomic structure and is porous. BC can be activated by physical or chemical methods to enhance its electron transfer rate, specific surface area, functional groups, and other properties [21,22]. ZnO is a non-toxic and environmentally friendly material with high biocompatibility [23]. We use the hydrothermal method to synthesize the ZnO/BC nanohybrid by doping it with biochar, which has a high electrical conductivity and could synergistically enhance the electrochemical detection properties. The morphology, phase composition, and purity of the synthesized materials were characterized and analyzed by scanning electron microscopy (SEM), X-ray diffraction pattern (XRD), and X-ray photoelectron spectroscopy (XPS). The electrochemical method was used to characterize the properties of the materials, and indicated that ZnO/BC had a better detection effect on BPA. In the detection of actual water samples, the accuracy of the ZnO/BC sensor was demonstrated by comparison with liquid chromatography.

## 2. Materials and Methods

### 2.1. Reagents and Apparatus

BPA, phenol, dichlorophenol (DCP), estradiol (E2), and ethinyl estradiol (EE2) were purchased from Sigma Aldrich (Shanghai, China). Zinc nitrate ($Zn(NO_3)_2$) disodium hydrogen phosphate ($Na_2HPO_4$), sodium dihydrogen phosphate ($NaH_2PO_4$), potassium chloride (KCl), sodium chloride (NaCl), potassium ferricyanide ($K_3Fe(CN)_6$), and potassium ferrocyanide ($K_4Fe[(CN)_6]$) were purchased from Sinopharm Chemical Reagent Co., Ltd. (Shanghai, China), all of which were analytically pure reagents. A stock solution of BPA was prepared in ethanol and was serially diluted with phosphate buffer (PBS, 100 mM, pH 7.0) to specific concentrations before electrochemical analyses. The whole experiment used a Merck's milli-Q water purifier (18.25 MΩ). Bagasse was purchased from a local market. Electrochemical analyses used an electrochemical workstation (CHI 660E) from Shanghai Chenhua Instruments, China. Biochar was prepared using a tube furnace (OTF—1200X, Hefeikejing Materials Technology Co., Ltd., Hefei, China), and was ground with a ball mill (YXQM—2 L, Changsha Mickey Instruments & Equipment Co., Ltd., Changsha, China). The powder XRD spectra were recorded by a Smartlab SEX-ray generator 3 kW closed tube (Rigaku, Beijing, China). SEM images were observed under an Xl-30 ESEM-FEG field emission scanning electron microscope (FEI, Hillsboro, OR, USA). The Oxford Instruments X-Max EDS system was used in conjunction with a scanning electron microscope to analyze

and determine the elemental composition of the materials. XPS was carried out on a PHI 5000 Versa Probe system (Ulvac-Phi, Chigasaki, Japan).

## 2.2. Preparation of ZnO/BC Nanohybrid

The sugarcane bagasse was washed with ultrapure water 3 times, dried in an oven, and put into the tube furnace. The program was set to raise the temperature by 7 °C/min in a $N_2$ atmosphere at 900 °C for 2 h and then decrease the temperature at 5 °C/min. The produced biochar was transferred to a ball mill for 1 h each in forward and reverse rotation at 300 rpm. Finally, a 100 mesh sieve was used to filter the ball-milled biochar, and then the under-sieved product was collected for later use. Subsequently, 0.3 mg of biochar nanoparticles was added to ultrapure water (50 mL) containing zinc nitrate (0.1 mg), followed by adjusting the pH to 10.0 with continuous sonication for 20 min. After that, the mixture was transferred to the PTFE liner of the reaction kettle and put into an oven at 180 °C for 6 h and naturally cooled down to room temperature. After separation, the material were centrifuged and washed with ultrapure water and ethanol, then dried in an oven and ground in an agate mortar to homogenize it, and it was denoted as a ZnO/BC nanohybrid. The same method was used to prepare ZnO, only without the addition of BC nanoparticles.

## 2.3. Preparation of ZnO/BC Electrochemical Sensor

Before modification, the glassy carbon electrode was polished to mirror-clean with 0.05 micron alumina powder and chamois, then the ZnO/BC nanohybrid was prepared as a solution of 1 mg/mL in ultrapure water, and 0.1% hydrochloric acid (1 M) was added to increase the dispersibility. After sonication for 5 min, 6 μg ZnO/BC of solution was coated onto the clean glassy carbon electrode and dried under infrared light to obtain a ZnO/BC sensor. The temperature during drying process should not be too high. The prepared ZnO/BC sensor was stored at 4 °C for subsequent use. BC and ZnO modified sensors (BC sensor, ZnO sensor) were prepared using the same procedure.

## 2.4. Preparation of Actual Water Samples

The feasibility of using the ZnO/BC sensor for practical analyses was evaluated using tap water and groundwater (Changchun, China). The water samples were filtered through a 0.45 μm filter and then diluted 10-fold with PBS. Finally, the water samples were used to prepare BPA solutions of different concentrations, and the electrochemical response signals were measured using the ZnO/BC sensor to evaluate the potential for practical applications. Meanwhile, high performance liquid chromatography (HPLC) was used to compare the accuracy of the experimental results. HPLC was performed using a UV-Vis detector and a Zorbax SB C18 column with a detection wavelength of 280 nm, and the mobile phase was methanol and water (65:35, *v/v*) at a flow rate of 1 mL/min.

## 2.5. Electrochemical Measurements

Electrochemical detection was performed using a three-electrode system consisting of glassy carbon electrode (GCE), Ag/AgCl, and platinum wire as the working, reference and auxiliary electrodes, respectively. The electrochemical behaviors of the GCE, BC/GCE, ZnO/GCE, and ZnO/BC/GCE were tested using electrochemical impedance measurements (EIS), differential pulse voltammetry (DPV), and cyclic voltammetry (CV). EIS analysis was carried out in 5 mM $K_3[Fe(CN)]_6/K_4Fe[(CN)_6]$ (containing 0.1 M KCl) at an amplitude of 5 mV and applied frequency range of 1 Hz to 100 kHz. CV analysis was performed in 20 mL of 50 mM PBS (pH 7.0) containing a specific amount of BPA with a potential range of 0.2–0.8 V and a scan rate of 100 mV/s. DPV analysis was conducted in the same solution of CV with a potential range of 0.2–0.8 V, amplitude of 0.05 V, and pulse width of 0.05. Before every measurement, the electrode was immersed in the BPA solution for 3 min for enrichment, unless specifically noted.

## 3. Results and Discussion

### 3.1. Characterization

The morphology of ZnO/BC was studied by SEM analysis. As shown in Figure 1B,C itself is a sheet-like structure. After hydrothermal treatment, the edges and corners were reduced and tended to be smooth. ZnO particles have a cluster structure, and after hydrothermal treatment with BC, it can be seen that part of the cluster structure was adhered to the BC, but it is not obvious. To further demonstrate the synthesis of the ZnO/BC nanohybrid, it was characterized by EDS mapping (Figure 1D). Figure 1E–G correspond to C, O, and Zn elements, respectively. This indicated that ZnO was uniformly dispersed on the surface of BC. The presence of ZnO was also confirmed from the element distribution histogram (Figure S1). This suggested that the ZnO/BC was successfully synthesized. The average particle size of ZnO/BC,404 nm, was slightly smaller than the pristine BC (542 nm) ($n = 100$) (Figure S1).

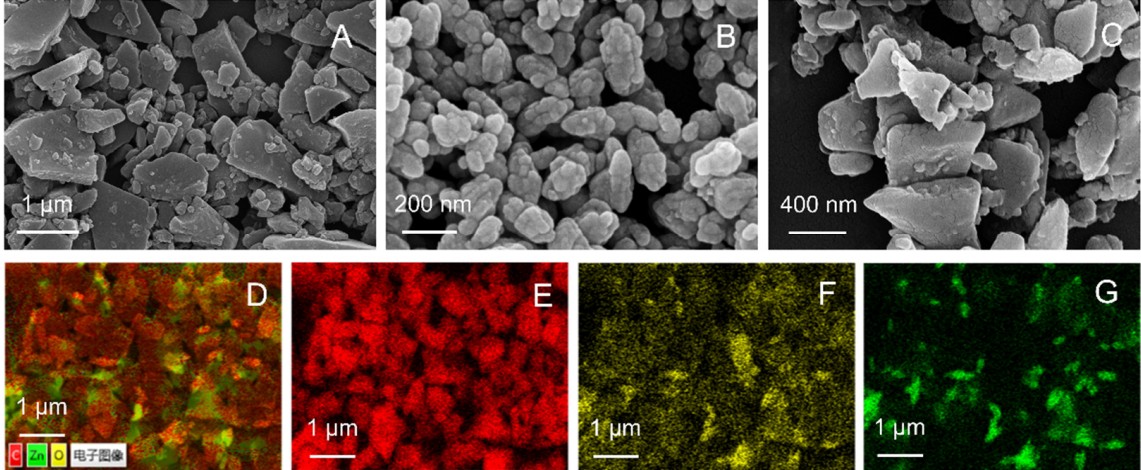

**Figure 1.** SEM images of (**A**) BC, (**B**) ZnO, (**C**) ZnO/BC. EDS layered image of (**D**) ZnO/BC (The electronic image). EDS mapping of (**E**) C, (**F**) O, and (**G**) Zn elements.

Figure 2 shows the XRD patterns of BC, ZnO, and ZnO/BC. The two slightly elevated diffraction peaks of BC at 23° and 43° belong to (002) and (100) refractions of the graphite planes [24]. The characteristic peaks of ZnO can all be found from in the spectra of ZnO/BC nanohybrid (PDF#89-0511) [25]. The diffraction peaks corresponding to BC and ZnO can also be found in the XRD patterns of the ZnO/BC nanohybrid. This confirms the successful synthesis of the ZnO/BC nanohybrid. To further demonstrate the elemental composition of the nanomaterial surface, XPS was performed to characterize the ZnO/BC. From the full XPS spectra of the ZnO/BC nanohybrid in Figure 3A, all the elements contained in the nanohybrid can be seen. It was proved that the synthesized material was pure. Figure 3B shows the C 1 s spectra at 284.7 eV, 286.2 eV, and 288.8 eV corresponding to C-C, C-O and C=O bonds, respectively. Figure 3C shows the O 1 s spectra at 531 eV, 531.62 eV and 532.48 eV corresponding to metal oxygen (ZnO), C=O and C-O/OH. Figure 3D shows the Zn 2p spectrum at 1022 eV and 1045.1 eV corresponding to $2p^{3/2}$ and $2p^{1/2}$ orbitals, respectively [26,27]. These data by XPS also illustrates the successful synthesis of the ZnO/BC nanohybrid.

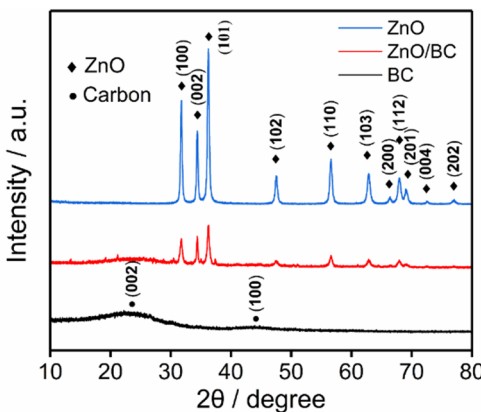

**Figure 2.** XRD patterns of ZnO/BC nanohybrid.

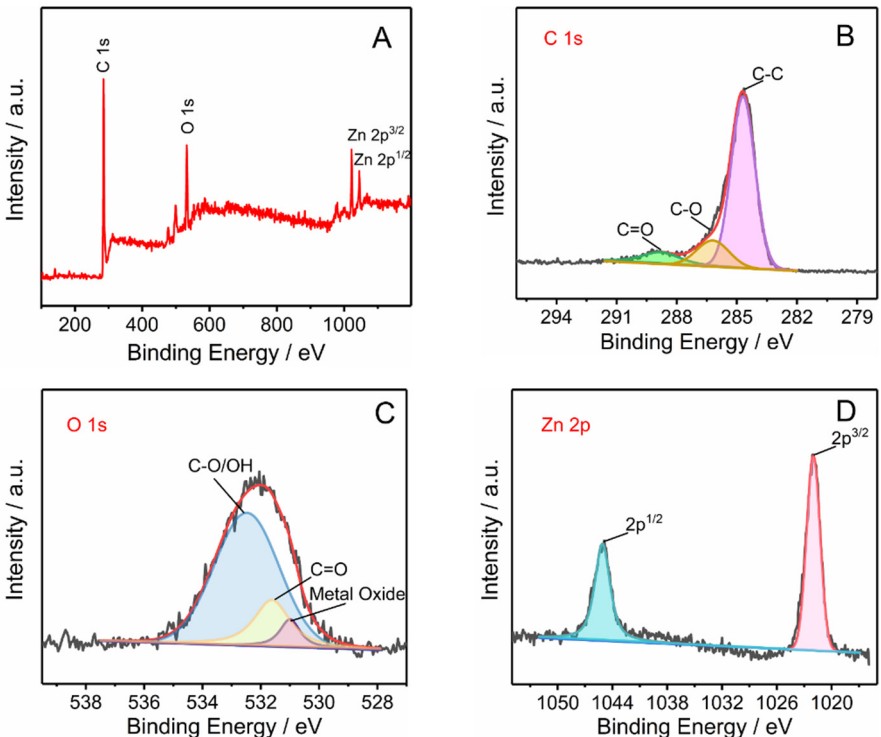

**Figure 3.** XPS survey spectrum of (**A**) ZnO/BC nanohybrid, core level spectra of (**B**) C 1 s, (**C**) O1 s, (**D**) Zn 2p.

As shown in Figure 4A, the Nyquist plots of the four modified electrodes were obtained from EIS in the presence of 5 mM $[Fe(CN)6]^{3-/4-}$ containing 0.1 M KCl. The resistance values ($R_{ct}$) of bare, ZnO, BC, and ZnO/BC electrodes were 65.82, 58.23, 34.83 and 15.9 $\Omega$ respectively, indicating that ZnO/BC had the fastest electron transfer ability which would facilitate the accelerated oxidation of BPA on the electrode surface. The DPV diagram for the determination of BPA in Figure 4B showed that the electrochemical signal by ZnO/BC (8.181 µA) was larger than GCE (5.068 µA), BC (6.94 µA), and ZnO (6.055 µA). This suggested that the electrochemical response of ZnO/BC was stronger due to the synergistic effect of ZnO and BC, which is beneficial for the detection of BPA. The CV also demonstrated that the ZnO/BC had a better performance for the detection of BPA (Figure S2). This should also be attributed to the defects caused by the generation of oxygen vacancies in ZnO and the presence of defects in metal oxides that would create a deep energy level to capture electrons and lead to the production of a large number of freely movable electrons, and enhance the electron transfer rate of the material [28].

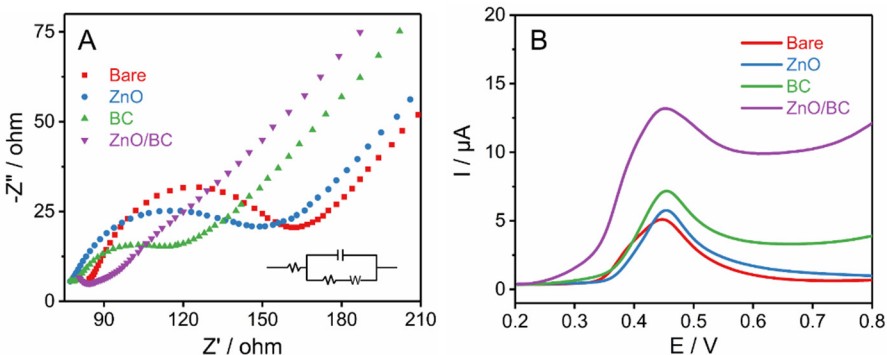

**Figure 4.** (**A**) EIS of modified electrode in 5 mM $K_3[Fe(CN)]_6/K_4Fe[(CN)_6]$ and 0.1 M KCl. (**B**) Comparison of DPV responses of modified electrode in PBS solution (50 mM, pH 7.0) containing 100 μM BPA with a scan rate of 100 mV/s (accumulation time of 5 min).

### 3.2. Experimental Parameter Optimization

Firstly, the synthesis conditions of the ZnO/BC nanohybrid, such as the synthesis time, material ratio, pH, and temperature, were optimized. As shown in Figure S3, the optimal BPA detection signal was found using the ZnO/BC nanohybrid synthesized at conditions of 180 °C, 6 h, pH 9.0, and with a mass ratio of 1:1 ($BC:Zn(NO_3)_2$), and were used in subsequent experiments.

The amount of material for electrode modification is one of the most important factors affecting the determination of BPA. Excessive modification will thicken the coating, resulting in increased resistance and a slower electron transfer rate [29]; thus, the performance of the sensor with various coating masses on the electrode surface was investigated, and it was found that the optimum oxidation current in response to BPA was with 5 μg of ZnO/BC (Figure S4). The adsorption performance of the material has a great influence on the detection time. Fast adsorption can improve the detection speed and decrease the detection limit to a certain extent. The electrochemical detection of BPA was carried out using ZnO/BC sensor with the adsorption time of 0 to 600 s. As shown in Figure 5A, detection signal increased rapidly at in the beginning stages, and reached a static value in later ones. Considering the time benefit, 180 s was used for subsequent experimental analysis. Under the optimal accumulation time, the detection performances of modified electrodes with different materials were compared (Figure S5), and the detection signal of ZnO/BC for BPA was the largest (9.345 μA), which indicated that ZnO/BC had a good improvement effect on the detection of BPA. This was due to the fast electron transfer rate and the easy accumulation of BPA by the hydrogen bonding, facilitated by the increased hydrogen oxide of ZnO/BC, as shown in Figure 5B,C. When the pH was in the range of 5.0–9.0, the peak potential shifted negatively toward the X-axis with the increase in pH, indicating that the proton was involved in the reaction at the electrode surface [28]. The electrochemical signal of the ZnO/BC sensor in response to BPA reached its maximum at pH 6.0 and then gradually decreased. There are multiple forces, such as electrostatic adsorption, π-π bonding, and hydrogen bonding, between the BPA and ZnO/BC electrodes. When the pH of the solution was greater than 4, the -COOH and -OH on the ZnO/BC would dissociate and be negatively charged. Considering BPA exists in a protonated form at pH 5.5–8.5 [29,30], as the pH is 5.0, BPA is not charged, and the force between BPA and the material should be relatively weaker than at the pH of 6.0. With the increase of pH, the hydrophobicity of BPA would increase, and the binding force of ZnO/BC to BPA may gradually be weakened. In addition, in alkaline conditions, the $OH^-$ would bind to protonated BPA and interfere with the interaction between BPA and ZnO/BC. Therefore, pH 6.0 was used for subsequent experimental analyses.

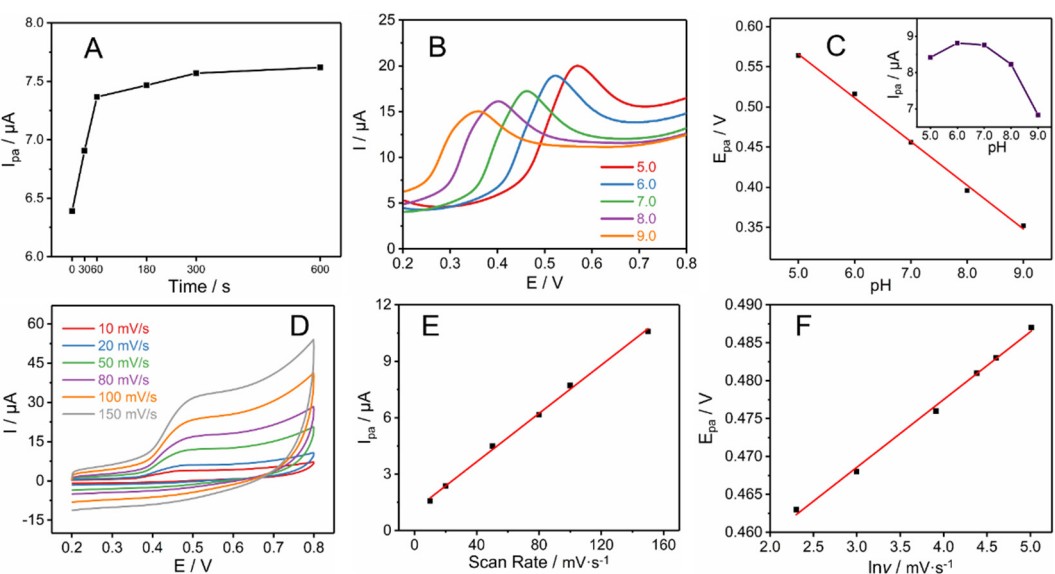

**Figure 5.** (**A**) Effect of adsorption time; (**B**) differential pulse voltammograms at different pH values; (**C**) point line graph of oxidation peak current ($I_{pa}$) vary with pH and the variation of peak potential ($E_{pa}$) with pH. (**D**) Cyclic voltammograms of ZnO/BC sensor at different scan rates, (**E**) variation of $I_{pa}$ with the scan rate ($v$), (**F**) variation of $E_{pa}$ with the natural logarithm of scan rate (ln$v$).

### 3.3. Mechanism Study of ZnO/BC Sensor for BPA Detection

The cyclic voltammograms of the ZnO/BC sensor at different scan rates (10–150 mV/s) are illustrated in Figure 5D. The results showed a linear relation between the anode peak current ($I_{pa}$) and scan rate ($v$) ($I_{pa} = 0.064\ v + 1.092$, $R^2 = 0.998$) (Figure 5E), which indicated that BPA was adsorbed onto the electrode surface during the detection process. Furthermore, for this irreversible reaction, the peak potential ($E_{pa}$) and the natural log of the scan rates [ln$v$] are linearly correlated ($E_{pa} = 0.0111\ \ln v + 0.512$, $R^2 = 0.998$) (Figure 5F). Based on the Laviron's equation [31]:

$$E_{pa} = E^0 + \frac{RT}{\alpha nF} \ln \frac{RTk_s}{\alpha nF} + \frac{RT}{\alpha nF} \ln v$$

where $E^0$ refers to the formal redox potential, R is the gas constant (R = 8.314), T is the absolute temperature (T = 298 K), $\alpha$ is the transfer coefficient and ranges between 0 and 1 for irreversible reaction processes, F is the Faraday constant (F = 96485 C mol$^{-1}$), and $k_s$ is the reaction kinetic constant, the electron transfer number (n) was approximately 2, which indicated that two electrons were involved in the redox reaction of BPA on the ZnO/BC electrode.

As observed from Figure 5C, the peak currents of BPA in ZnO/BC electrode showed pH dependent behavior. The $I_{pa}$ and $E_{pa}$ were affected by pH changes. The $E_{pa}$ and pH of ZnO/BC were linearly correlated ($E_{pa} = -0.054\ \mathrm{pH} + 0.838$, $R^2 = 0.997$) (Figure 5C). The slope of the equation was $-54.4$ mV/pH, which was close to the theoretical value of $-59.2$ mV/pH, indicating that the number of electrons and protons involved in the oxidation reaction of BPA on the electrode surface were equal [32]. Thus, the oxidation reaction process on the surface of the ZnO/BC electrode involved two electrons and two protons, which was consistent with that reported in the previous literature [33]. Combining the above, we speculate that the oxidation mechanism of BPA on ZnO/BC electrode is as shown in Scheme 1.

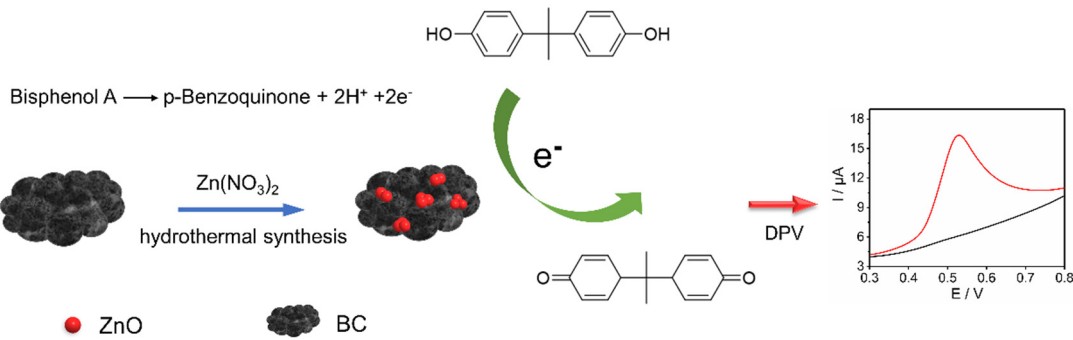

**Scheme 1.** The oxidation mechanism of BPA on ZnO/BC sensor.

### 3.4. Electrochemical Detection of BPA

The electrochemical detection performance of ZnO/BC sensor for BPA was investigated by DPV. As shown in Figure 6A, the oxidation peak current varies with the concentration of BPA, and the oxidation peak currents were linearly related to the concentration of BPA in the range of $5 \times 10^{-7} \sim 1 \times 10^{-4}$ mol/L, with linear equations of $I_{pa}$ ($\mu$A) $= 0.092\, C$ ($\mu$M) $+ 0.09$ ($R^2 = 0.993$) (Figure 6B). The detection sensitivity was 92 mA/M, and the limit of detection was $1 \times 10^{-7}$ mol/L (S/N = 3). As compared with previous reports, the ZnO/BC sensor presented a comparable linear range and detection limit (Table 1). This detection method also has advantages, such as less consumption of materials and rapid detection.

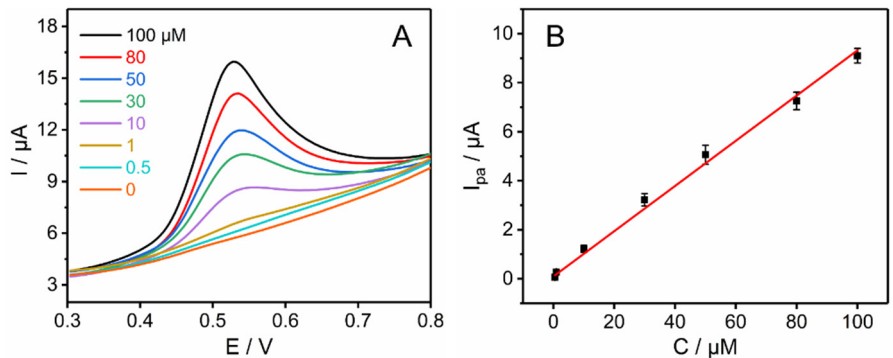

**Figure 6.** (**A**) DPV responses of the ZnO/BC sensor in different concentrations of BPA (50 mM PBS, pH 6.0). (**B**) Calibration curve for the determination of BPA (0–100 $\mu$M).

**Table 1.** Comparison with some previous sensors for BPA detection.

| Sensor [1] | Linear Range ($\mu$M) | Detection Limit ($\mu$M) | References |
|---|---|---|---|
| PEDOT/GCE | 90–410 and 40–410 | 50 and 22 | [34] |
| Au/ssDNA/SWCNT | 0.5–3.8 | 0.011 | [8] |
| Au/Gr-AgCu | 0.1–100 | 1.31 | [35] |
| SiO2/GO/AgNP/GCE | 0.1–2.6 | 0.045 | [18] |
| SWCNT/GCE | 10–100 | 7.3 | [36] |
| RGO-Ag/PLL/GCE | 1–80 | 0.54 | [15] |
| CTpPa-2/GCE | 0.1–50 | 0.02 | [28] |
| ZnO/BC/GCE | 0.5–100 | 0.1 | This work |

[1] PEDOT: poly(3,4-ethylenedioxythiophene), Au: gold electrode, SWCNT: single-walled carbon nanotubes, Gr-AgCu: graphene-bimetallic nanoparticle composites, RGO-Ag/PLL: reduced graphene oxide-silver/poly-L-lysine, CTpPa: a porous crystalline covalent organic framework.

### 3.5. Reproducibility, Stability, Anti-Interference and Practical Application

A good electrochemical sensor must have repeatability, stability, and anti-interference ability in order to be suitable for the complex situation of environmental pollution. To evaluate the reproducibility of the sensor, we used six independent glassy carbon electrodes to prepare ZnO/BC electrodes, and then detected the same concentration of BPA (20 μM) solution. As shown in Figure 7A, the electrical signals of the six independent electrodes were almost unchanged when detecting BPA, and the relative standard deviation (RSD) was calculated to be 4.3%, indicating that the ZnO/BC sensor had good reproducibility. To evaluate the stability of the sensor, it was stored at a low temperature of 4 °C. As shown in Figure S6, the electrochemical detection of BPA did not change significantly within one week, indicating that the sensor has high stability. It is modified using the drop coating method, so after the ZnO/BC solution is prepared it is not only extremely convenient to use, but also can be used many times. It only took 5 min to prepare a complete ZnO/BC electrode, and the solution was stored at 4 °C for 2 months without a signal drop.

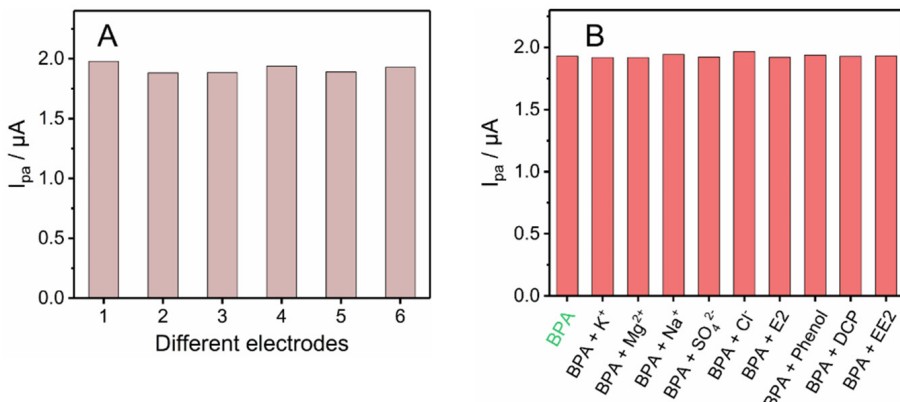

**Figure 7.** (**A**) Reproducibility of ZnO/BC sensor for 20 μM BPA detection. (**B**) Oxidation peak currents of 20 μM BPA at the ZnO/BC electrode in the presence of 100-fold anions, cations and the same concentration of estradiol (E2), phenol, dichlorophenol (DCP) and ethinylestradiol (EE2).

The anti-interference ability of the ZnO/BC sensor was evaluated using several traditional ionic and organic pollutants. As shown in Figure 7B, the effects of $Na^+$, $K^+$, $Mg^{2+}$, $Cl^-$, $SO_4^{2-}$ ions were negligible, even if the concentration was 100-fold higher than the BPA. For organic pollutants, such as E2, phenol, DCP, and EE2, the sensor could still specifically recognize BPA at lower concentrations. However, as the concentration of the interfering substances rose to a certain level, there was some interference (Figure S7), probably due to the competitive accumulation on the electrode. The selectivity was possibly improved through variation of the material's functional groups and buffer pH, to enable preferential adsorption of target molecules.

The practical application ability of the ZnO/BC sensor in the environment was evaluated by the spike recovery method (Table 2). According to the pollution status in the contaminated site [37], three concentrations of BPA (1, 10, and 100 μM) were added to groundwater and tap water samples, and then detected using the ZnO/BC sensor. Satisfactory results were obtained. At the same time, in order to verify the accuracy of the experimental results, standard samples were detected and analyzed using HPLC. The experimental results obtained by HPLC were consistent with the ZnO/BC sensor. Therefore, the ZnO/BC sensor can be applied in actual sample analyses.

**Table 2.** Determination of BPA in real water with the ZnO/BC sensor and HPLC.

| Samples | | Detected by the ZnO/BC Sensor | | | Detected by HPLC | |
|---|---|---|---|---|---|---|
| | Added (µM) | Average found [1] (µM) | Recovery (%) | RSD (%) | Average found [1] (µM) | Recovery (%) |
| Groundwater | 1.0 | 0.96 | 96.00 | 3.92 | 0.95 | 95.00 |
| | 10.0 | 10.04 | 103.70 | 4.67 | 10.28 | 102.80 |
| | 100.0 | 100.06 | 100.06 | 5.41 | 99.01 | 99.01 |
| Tap water | 1.0 | 1.02 | 102.00 | 2.94 | 0.99 | 99.15 |
| | 10.0 | 9.92 | 99.17 | 3.28 | 9.70 | 96.99 |
| | 100.0 | 99.55 | 99.55 | 1.44 | 95.91 | 95.91 |

[1] Average of 3 measurements.

## 4. Conclusions

A highly conductive ZnO/BC nanohybrid was successfully synthesized and applied to the detection of BPA in water for the first time. Accurate identification of BPA in water using the ZnO/BC sensor took only 3 min. The ZnO/BC sensor can be fabricated quickly and has a long storage stability. In addition, the preparation of modified electrodes is simple and the raw materials are widely available. Compared with HPLC, it was further proved that the ZnO/BC sensor can accurately measure the actual water samples. Further, ZnO is a non-toxic material that can be used in external medicine, so it has no impact on health and safety. Therefore, this cost-effective synthesis of ZnO/BC nanohybrid has important value in the preparation of stable and low-cost sensors.

**Supplementary Materials:** The following supporting information can be downloaded at: https://www.mdpi.com/article/10.3390/chemosensors10050163/s1, Figure S1: (A) Total elemental distribution spectrum and (B) the particle size distribution histogram of BC and ZnO/BC; Figure S2: CV responses of modified electrode in PBS solution (50 mM, pH 7.0) containing 100 µM BPA with a scan rate of 100 mV/s; Figure S3: Effect of hydrothermal synthesis of ZnO/BC (A) time, (B) proportion, (C) pH and (D) temperature on the amperometric responses to 100 µM BPA in PBS solution (50 mM, pH 7.0); Figure S4: Histogram of the effect of drop mass on the response signal; Figure S5: Differential pulse voltammogram for comparison of adsorption performance (A), Histogram of peak current values from the detection of 100 µM BPA using different materials (B); Figure S6: Stability of the ZnO/BC sensor stored at 4 °C; Figure S7: Differential pulse voltammogram at interferer concentrations 5 times higher than BPA (20 µM) (A), corresponding peak-current histogram (B) [38].

**Author Contributions:** J.H.: Investigation, Experiment, Writing—Original Draft. D.M.: Writing—Review and Editing. P.D.: Resources. K.L.: Investigation. Y.L.: Investigation. X.W.: Resources. Y.P.: Supervision, Writing—Review and Editing. All authors have read and agreed to the published version of the manuscript.

**Funding:** This research was funded by the National Key Research and Development Program of China (2019YFC1804800) and the National Natural Science Foundation of China (51809111).

**Institutional Review Board Statement:** Not applicable.

**Informed Consent Statement:** Not applicable.

**Data Availability Statement:** Not applicable.

**Acknowledgments:** Thanks to the Key Laboratory of Groundwater Resources and Environment of the Ministry of Education of Jilin University for providing the instrumental and technical support for the high-performance liquid chromatography analysis.

**Conflicts of Interest:** The authors declare no conflict of interest.

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
