# Peer review of "Green Synthesis of ZnO/BC Nanohybrid for Fast and Sensitive Detection of Bisphenol A in Water"

_chemosensors, doi:10.3390/chemosensors10050163_

Round 1
Reviewer 1 Report
The article is interesting and data are coherent, however the language must absolutely be improved.
I strongly suggest reviewing the article with the help of a native speaker
Beside this, I have the following issues:
-Scheme 1: replace the “molecular model” formulas of BPA and chloranil with classically written formulas (using text, not grey/black and white spheres)
-Table 2: please check for the correct number of significant figures
-Pag 6 lines 218 and following: the explanation of the reason for which the optimum working pH is 6 is not clear, as according to it, the sensor should work better at pH 5 or even less. Moreover, being the pKa 9.73, there should be no difference between pH 8, 7 or 6.
-The authors should explain better the reason for working with a modified electrodes, as the determination of BPA is possible also with an unmodified screen printed electrode (Chemosensors 2020, 8, 103; doi:10.3390/chemosensors8040103)
-I suggest to the authors to try as possible interference hydroquinone and catechol, as done by other authors, as they are similar to BPA and can be present in real samples. I don’t see the reason for using EES2 at these high concentration as a possible interfering compound
-p 8 line 293 and following: I think it is difficult to recognise the presence of the interferences by looking at a peak shape change in intermediate cases. Please reformulate the sentence. What is the concentration of BPA used to obtain Fig S7?
-Please specify in the different captions of the figure what is the accumulation time used (I think 180 s?) and specify in the text if it is performed at open circuit and/or under forced convection.
-p7 eq line 236. Have you used alpha = 0.5? Otherwise, from your calculation you can’t obtain the number of electron involved in the electrochemical reaction, but just the product of alpha and n
-p 7 line 231: the behaviour suggest that adsorption take place. What do you mean exactly with “kinetically controlled electrochemical process”? Diffusion is not involved at all in the process. Fig 5D is obtained with pre-concentration?
- tab 2: 1 microM BPA means around 230 microg/L. Is this a reasonable concentration found in polluted waters? Can you please provide any reference at this regard?
-detection sensitivity is reported in the abstract but not in the text
Reviewer 2 Report
The paper "Green synthesis of ZnO/BC nanohybrid for fast and sensitive detection of bisphenol A in water" presents a very relevant method to detect substances in water. It successfully demonstrate how electrochemical sensors can be applied for detection of specific substances in water, and also using green non-toxic materials as sensing materials. The paper is well written and I would recommend it for publication in Chemosensors after some minor revisions:
- I am missing a better introduction to BC, what is that, how is extracted, and why when combined with ZnO, can selectively bind to BPA?
- It is a bit difficult to read the scale bars in figure 1. Also, from which of the images is the EDS measurements extracted. This could be improved.
- How could one improve the selectivity of the sensor to BPA when similar substances are present? Some outlook could be presented.
- Lines 262-263, use of the expression "On the other hand" twice can bring some confusion.
Round 2
Reviewer 1 Report
The authors have adequately answered to all the questions raised by the reviewers, and the manuscript can be published.
Author Response
Dear Reviewer,
Thank you very much for your comments on this paper.